# Bespoke Implants for Cranial Reconstructions: Preoperative to Postoperative Surgery Management System

**DOI:** 10.3390/bioengineering10050544

**Published:** 2023-04-29

**Authors:** Mihaela-Elena Ulmeanu, Ileana Mariana Mateș, Cristian-Vasile Doicin, Marian Mitrică, Vasile Alin Chirteș, Georgian Ciobotaru, Augustin Semenescu

**Affiliations:** 1Faculty of Industrial Engineering and Robotics, University POLITEHNICA of Bucharest, 060042 Bucharest, Romania; mihaela.ulmeanu@upb.ro; 2Central Military Emergency University Hospital “Dr. Carol Davila”, 010825 Bucharest, Romania; titimitrica@yahoo.com (M.M.); achirtes@yahoo.com (V.A.C.); neurochirurgie@doctorciobotaru.ro (G.C.); 3Faculty of Materials Science and Engineering, University POLITEHNICA of Bucharest, 060042 Bucharest, Romania; augustin.semenescu@upb.ro; 4Academy of Romanian Scientists, 3 Ilfov St., 050044 Bucharest, Romania

**Keywords:** cranioplasty, bespoke cranial implant, additive manufacturing, surgery management system

## Abstract

Traumatic brain injury is a leading cause of death and disability worldwide, with nearly 90% of the deaths coming from low- and middle-income countries. Severe cases of brain injury often require a craniectomy, succeeded by cranioplasty surgery to restore the integrity of the skull for both cerebral protection and cosmetic purposes. The current paper proposes a study on developing and implementing an integrative surgery management system for cranial reconstructions using bespoke implants as an accessible and cost-effective solution. Bespoke cranial implants were designed for three patients and subsequent cranioplasties were performed. Overall dimensional accuracy was evaluated on all three axes and surface roughness was measured with a minimum value of 2.209 μm for Ra on the convex and concave surfaces of the 3D-printed prototype implants. Improvements in patient compliance and quality of life were reported in postoperative evaluations of all patients involved in the study. No complications were registered from both short-term and long-term monitoring. Material and processing costs were lower compared to a metal 3D-printed implants through the usage of readily available tools and materials, such as standardized and regulated bone cement materials, for the manufacturing of the final bespoke cranial implants. Intraoperative times were reduced through the pre-planning management stages, leading to a better implant fit and overall patient satisfaction.

## 1. Introduction

According to statistics, traumatic brain injury (TBI) is a leading cause of death and disability worldwide, with an estimated 69 million cases each year [1]. Severe TBI accounts for 8% (about 5.48 million people) of the total registered cases, requiring surgical intervention [2]. According to the World Health Organization (WHO), low- and middle-income countries, where 85% of the population resides, account for nearly 90% of deaths resulting from injuries. TBI is the primary cause of one-third to one-half of these trauma-related deaths and is the leading cause of death and disability worldwide among all trauma-related injuries [3]. The management of traumatic brain injuries can be a challenging and costly process, with no guarantees in the recovery outcome [4]. The burden of disease could be significantly reduced by developing clinical practice guidelines and through increased international collaboration on good practices [5]. In severe cases of brain injuries, a craniectomy may be necessary to relieve pressure on the brain and prevent further damage [6]. During the procedure, a portion of the skull is removed and stored, and the brain is allowed to expand without being compressed by the skull. Apart from trauma, cranial defects or deformities can also occur as a result of infection, congenital malformations, or abnormal growths, such as tumors [7,8,9,10]. In these cases, a cranioplasty may be necessary to repair or replace the missing portion of the skull and restore normal brain function. Craniectomy and cranioplasty are often performed together as complementary procedures [11]. After the relief of intracranial pressure via craniectomy, the cranioplasty is conducted to restore the integrity of the skull for both cerebral protection and cosmetic purposes. The restoration of the skull through cranioplasty may also help to re-establish cerebral blood flow and cerebrospinal fluid dynamics, thus contributing to neurological recovery [12,13]. There are several methods to perform a cranioplasty, and the choice of technique will depend on the specific needs of the patient, the size and location of the cranial defect, and the surgeon’s experience and preference [14]. Some of the most common methods for performing a cranioplasty are autograft, allograft, computer-aided design, and the manufacturing (CAD/CAM) of implants or custom-made implants.

The autograft method involves using the patient’s own bone to repair the defect. The surgeon will typically conserve the cranial bone after performing a craniectomy and later use it to reconstruct the skull in a subsequent surgical procedure [15]. For the allograft method, the surgeon uses a donor bone or synthetic material to repair the defect [16]. The allograft may be obtained from a cadaver or a bone bank. The CAD/CAM method involves creating a 3D model of the patient’s skull defect using imaging scans, such as CT or MRI [17]. The surgeon can then use this model to create a custom-made implant or guide the placement of an autograft. Once the tissue has grown enough to fill the defect, a cranioplasty procedure can be performed. Custom-made implants are usually made of titanium or other materials and are designed to fit the specific shape of the patient’s skull defect. The implant is typically created using a 3D-printing process, and the surgeon will place it over the defect and secure it in place with screws or other hardware [18]. Vacuum-assisted closure (VAC) therapy is sometimes used in speeding up the recovery process of cranial wounds and can promote healing and soft tissue regeneration [19].

Autograft and custom-made implants are generally considered to be the most commonly used methods for cranioplasty [16]. Autograft is preferred by many surgeons because it reduces the risk of rejection or infection, which is associated with using foreign materials [20]. The bone graft can be taken from another part of the patient’s body, such as the rib or hip, and then shaped and secured into place over the defect. There are some major drawbacks to autografts, mainly related to the limitations in shape and size, variable quality, availability of viable bone tissue, and donor site morbidity [21]. Autografts are limited in their ability to achieve complex shapes or large sizes and the quality of the bone used for the graft can vary depending on factors, such as the patient’s age, medical history, and overall bone health. In some cases, there may not be enough suitable bone available for an autograft, especially if the defect is large or if the patient has a medical condition that affects bone quality or quantity. These issues can affect the success and durability of the cranioplasty and usually lead to using bespoke cranial implants. They are designed to fit the specific size and shape of the patient’s skull defect [22,23]. Advanced imaging and CAD technologies are used to create a 3D model of the patient’s skull defect and then produce an implant that precisely matches the shape of the defect.

There are a variety of methods and tools used to develop custom implants for cranial reconstructions [22]. One of the most common approaches is to use CAD applications and additive manufacturing (AM) to design and manufacture bespoke implants [24]. AM has been shown to be an accurate and effective method for producing custom-made implants, with good patient outcomes reported in several studies. Custom-made implants can be made from a variety of materials and can provide a good cosmetic outcome [25,26,27]. Bespoke cranial implants are often manufactured from titanium (Ti), polyether-ether-ketone (PEEK), polymethylmethacrylate (PMMA), hydroxyapatite (HA), or polyurethane (PU) [28]. Studies have shown that complications are statistically significantly higher for autologous bone compared to combined alloplasts, such as HA, PMMA, and Ti [11,20]. The choice of material for a custom cranial implant depends on factors, such as the size and location of the defect, the patient’s medical history, the surgeon’s preference and experience, and infrastructure availability.

A study published in 2021 in the World Neurosurgery Journal by Nguyen et al. [23] reported on the use of 3D printing to create custom-made titanium implants for 35 patients with skull defects, with good results reported in terms of implant fit, stability, and aesthetic outcomes. Reported complications included four postoperative hematomas and one surgical site infection. Research conducted by Sharma et al. [29] showed that AM was used to design and produce biomimetic implants with excellent cosmetic and functional outcomes. The design of the lightweight cranial prosthesis made of titanium was evaluated using Selective Laser Melting (SLM) technology and presented high-dimensional accuracy. Designed implants were also supplied with orifices to efficiently prevent extradural hematomas. A limitation of the study was the lack of in vivo testing.

Pöppe et al. [30] report on using PMMA cranial implants manufactured using custom made 3D-printed templates with the springform technique in cranioplasty surgeries for 14 patients. No intraoperative complications were recorded and the molding of the PMMA material went smoothly, resulting in implants that fit well into the craniectomy defects. Subsequent CT scans showed excellent reconstruction of the skull in all cases. However, three patients with known risk factors for postoperative hematoma required revision surgery due to epidural hematoma. No patients experienced any new or permanent neurological deficits or died as a result of the surgery. Customized PMMA implants have also been obtained using 3D-printed polylactic acid molds as Hay et al. [31] describe in a technique applied for two cranioplasty patients. Excellent cosmetic results were obtained, and postoperative CT scans indicated restoration of the symmetrical contours of the cranium. Neither patient experienced any neurological or infectious complications during a 6-month follow-up. Low-cost PMMA implant manufacturing techniques involve manual shaping of the material on top of a 3D-printed patient skull [32]. Singh DK et al. conducted an observational, retrospective, and cohort study that included 20 patients on which a PMMA-based cranioplasty flap was used. No evidence of bleeding, infection, or poor scar formation was observed throughout the study. Limitations included the accuracy of the implant, which is highly dependent on the technician or surgeon’s skill set when the manual shaping is conducted.

In research conducted by Wandell and his team, the surgical precision of PEEK cranial implant insertion was assessed by contrasting a plan based on computer tomography (CT) with the actual postoperative position in twelve patients [33]. The main findings showed that the root mean square error between the planned position and the actual position of the implant ranged from 0.66 mm to 3.1 mm. This degree of precision indicates that PEEK implants can be positioned with sufficient accuracy to attain satisfactory patient aesthetics. Further research is needed to evaluate the functionality of their placement method. Other studies have shown that 3D-printed PEEK implants have good load-bearing capacity [34], registering a maximal Von Mises stress of 8.15 MPa, Von Mises strain of 0.002, and deformation of 0.18 mm in FEA ANSYS.

Most of the research and findings in this area address, separately or as a combination, the following topics: medical imaging processing, CAD design process, manufacturing stages, and intraoperative procedures [35]. Thus, an overview of the entire process is difficult to put together. Additionally, long-term patient follow-up usually does not involve a correlation between the patient’s outcome and the postoperative and intraoperative surgical strategies used [36]. Other limitations of bespoke cranial implants refer to high costs, longer wait times, material limitations, infection risks, imaging limitations, and the potential need for revision surgery [37]. An operation time exceeding 90 min, early cranioplasty, patient age over 20, and female gender are among several infection risk factors [38].

For overall benefit and patient compliance, the authors propose an integrative surgery management system for developing bespoke implants used in cranial reconstructions. The work aims to streamline surgical procedures that use bespoke implants in oral and craniofacial reconstruction surgeries, with the final target of improving patient outcomes and wellbeing through the reduction of the main limitations of bespoke cranial implants. Costs are lowered through the use of readily available tools and materials in the operating theatre and a regular engineering workshop. Standardized and regulated bone cement materials are used for the manufacturing of custom implants, thus limiting infection risks and the potential need for revision surgery. Intraoperative times are reduced through the pre-planning management stages, leading to a better implant fit and patient compliance, reduced post-operative complications, and reduced surgeon fatigue.

## 2. Materials and Methods

An integrative surgery management system (ISMS) was designed with three main components, namely preoperative, intraoperative, and postoperative (Figure 1). The ISMS was used on three patients (two males and one female) between September 2020 and March 2023, with no revision surgeries necessary so far. In the first stage of the preoperative component, medical data was collected at the Central Military Emergency University Hospital “Dr. Carol Davila” through computer tomography (CT) scans. Clinical evaluation revealed all three patients were good candidates for cranial reconstruction using customized implants.

Data processing was undertaken using the specialized Mimics software (v18.0), which allowed the transformation of DICOM images into STL files. A 3D model of the skull with the cranial defect was the main source for the reconstruction of the anatomical bespoke cranial implant. This step was achieved using the 3-matic software (v9.0). The collection of medical data from patients was done using CT scans (Table 1).

The CT characteristics were further used to import and process the DICOM image sets for each of the three patients. Each set of images was captured at a minimum slice thickness of 0.625 mm and a maximum of 1.25 mm, whilst pixel spacing was set between 0.40 mm and 0.55 mm.

Next, the prototype implant and the skull cranial defect were manufactured using an FDM Zortrax M300 Plus 3D printer fitted with a 0.4 mm nozzle, taking into consideration the optimization of 3D printing parameters. After selection of the optimum 3D printing profiles for all parts, machine maintenance had to be performed in preparation for the 3D-printing process. Automatic print bed leveling was undertaken before all prints. Travel axes were cleaned and degreased for the 4-day long prints. For the skulls, which weighed over 800 g, the maximum quantity available for the 3D-printing filament spool and the pause protocol were activated. A sensor fitted on the machine identified when the spool only had 0.5 m of filament left and paused the print job. During the pause, the equipment purges the existing material and loads a new spool of filament. After the material was changed, the skull print job was resumed.

After 3D printing was complete, parts were left approximately 30 min to cool down, as they had very thin wall structures. If the parts had been removed while the build platform was still hot, the general shape of the implants would have been deformed and proper assembly would not have been possible in the fit test phase. All 3D-printed parts were removed with a spatula from the build plate and post-processing operations were performed [39]. Rafts were deburred and support structures were removed using pliers.

Accuracy verification of the 3D-printed parts was undertaken for all three master part implants. Dimensional accuracy was evaluated using a Mitutoyo IP76 Caliper. Surface roughness was tested in 8 points on each of the three implants using an Insize C002 surface roughness tester. The first four points were set on two perpendicular axes on the outside convex surfaces and the next four points on similar axes placed on the inner concave surfaces of the implants.

Fit tests were undertaken using the 3D printed models and, if necessary, the implant was improved and redesigned until perfect assembly was achieved. Surgeons used the bio-models to plan the surgical procedure. After the surgeon’s validation, a custom mold was manufactured from bicomponent room temperature vulcanized (RTV) silicone rubber. The 3D-printed implant was used as the master part and set the separation plane. Mechanical characteristics of potential materials involved in the manufacturing of the final implant were analyzed (Table 2).

The 3D-printed bespoke implant was fixed in place in a cylindrical recipient half filled with modeling clay. A bicomponent ZA 22 liquid silicone rubber was selected for this application due to its characteristics. The chosen silicone was mixed in a 1:1 ratio base and catalyst and poured on top of the fixed 3D-printed bespoke implant modelling clay. It vulcanized at room temperature (RTV) in a minimum of 3 h. For a full cure, the mold sat at room temperature for 24 h or it could have been placed in an oven at 120 °C for 1.5 h. After one day, the modelling clay was removed, and the hardened silicone assembled with the 3D-printed implant was flipped over. Demolding agent was sprayed over the implant and exposed silicone surfaces. Mixed silicone was poured again to form the second part of the mold and was left to cure for another 24 h. After a full cure, the mold was released from the cylindrical recipient and the master part removed, revealing a cavity with the same shape of the implant. A PMMA implant was cast to test and validate the functionality of the manufactured mold.

The second intraoperative component started with surgical theatre preparation. Standard and hospital procedures and protocols were ensured. An EVO steam autoclave was used to sterilize the custom silicone mold for 30 min at 121 °C according to standard CDC guidelines.

Next, the patient was prepared for surgery and the cranial defect was exposed through the removal of skin and cicatricial tissue from the wound site. Standard hospital procedures and protocols were followed. All equipment and instruments that were used during the procedure were sterilized using the hospital-approved methods, such as autoclaving and chemical sterilization. A steam autoclave was used to sterilize the custom silicone mold for 30 min at 121 °C according to standard CDC guidelines. The silicone mold was thoroughly cleaned of any debris or biological material, using an ultrasonic cleaner. Once the silicone mold was cleaned, it was wrapped in sterilization packaging material to protect it during the sterilization process. It was placed in an autoclave, which was programmed to run a sterilization cycle that included heating the chamber to a high temperature and pressurizing it with steam. After the sterilization cycle was complete, the autoclave chamber was allowed to cool down before the silicone mold was removed. 

Stryker Antibiotic Simplex was used for in situ casting of the final custom implant (Figure 2) inside the mold cavity. The acrylic bone cement was selected due to its specific components and characteristics. It is important to mention that the selected bone cement is radiopaque and contains erythromycin and colistin used for antibacterial properties [40]. The main listed ingredients and content in percentages are as follows: poly(styrene-co-methyl methacrylate)—87 to 91%; barium sulphate—9 to 11%; erythromycin glucoheptonate 1 to 2%; and colistin methanesulphonic acid, sodium salt—<1%. The casting material was packaged in two sterile components. One component was an ampoule containing 20 mL of a colorless, flammable liquid monomer that has a sweet slightly acrid odor and contains methyl methacrylate (monomer), N,N-dimethyl pare toluidine, and hydroquinone. The other component was a packet of 41 g of finely divided powder containing methyl methacrylate–styrene copolymer, polymethyl methacrylate, barium sulphate USP and EP, erythromycin glucoheptonate USP and colistin sulphomate sodium EP. The content of the two components was mixed intraoperatively (Figure 2). The mixture resulted in the exothermic polymeric formation of a soft, pliable, dough-like mass, which, as the reaction progressed, became a hard cement-like complex. While still pliable, the mixture was placed inside the silicone mold and was pressed into shape while the exothermic reaction took place. Demolding of the final bespoke PMMA implants was done after approximately 5 min and extra material was deburred with an electric drill. 

The PMMA implant was positioned in relation to the patients’ cranial defect and the shape was validated. Edges were buffed with an electrical drill if perfect fit was not achieved from the first try. After shape and position validation the PMMA implant was fixated in place with titanium plates. The wound was sutured closed and the surgical site was sterilized.

The final postoperative component included a CT scan for confirmation of correct implant positioning and fixation. Visual aesthetics and patient compliance were also validated through pictures and compared with the natural curve of the patients’ skulls. Strict patient monitoring was undertaken while patients were hospitalized for implant compliance and further possible complications. Patients received continuous and constant care and follow-up after release from the hospital. Long-term recurrent monitoring was done using magnetic resonance imaging (MRI) scans.

## 3. Results and Discussions

### 3.1. Preoperative Actions

As per the ISMS, the preoperative stage started with the collection of medical data from patients using CT scans (Table 1). Various patients undertook all other medical investigations necessary to establish if they qualify for cranioplasty using custom-made implants. Only clinically healthy patients were selected for the procedure. All three selected patients for this study undertook the ISMS under the care of healthcare specialists from Central Military Emergency University Hospital “Dr. Carol Davila” from Bucharest.

The DICOM images captured using CT scans were transformed using Mimics software (v18.0) into STL files for processing. First, a mask was created for the entire skull structure with the patient’s cranial defect (Figure 3). This was done from the Project Management Masks tab by selecting the *New Mask* feature. A bone CT was set as the predefined threshold set with a minimum of 226 HU. The *Fill holes* and *Keep largest* options are selected. After mask generation, the Mask 3D Preview and 3D Navigation Indicator were activated from the 3D preview window. After these selections were performed, the active and selected masks were visible in all windows of the software. Next, the *Calculate Part* operation was applied on the skull mask with the optimal quality options activated. Parts were calculated differently for each patient. The part for Patient 1 had the first sliced positioned at 2.00 mm and the last slice at 181.10 mm. Patient 2 had the position of the first slice set at −78.970 mm and the position of the last slice at 84.829 mm. Matrix reduction in XY resolution was done with 0.4883 mm and in Z resolution with 0.2989 mm. For Patient 3, the position of the first slice was set at −88.125 mm and the position of the last slice at 104.375 mm. The matrix was reduced in XY resolution with a value of 0.5254 mm and in Z resolution with 1.25 mm. Further smoothing, reducing, or warping operations can be performed if the resulting object does not have an appropriate mesh for the following steps. The created object was exported as an STL file and was ready for the next step, namely, the reconstruction of anatomical bespoke cranial implant.

The first patient had an affected area with a measured perimeter of approximately 36.3 cm, the second patient a perimeter of 29.3 cm, and the third a perimeter of 23.5 cm.

Reconstruction was done by importing the STL skull file into the 3-matic software (v9.0). The first applied operation was *Fix* with the *Reduce* option at 0.5 and *Smooth* option at 0.7. An outside mesh of the skull was made using the *Wrap* operation from the Design tab of the software interface. A new part was thus created, and the initial skull mesh was hidden to avoid unwanted changes throughout the design process. Alongside the perimeter of the skull defect, a new 3D curve was defined using the *Create curve* operation with the following options activated: smooth curve, attach curve, attract curve, and close curve. A curvature analysis was undertaken to visually identify any areas in which the newly created curve might generate negative edges on the bespoke implant. The curvature analysis had a range of 0.5 to 0.25. If necessary, the curve was edited, to ensure tangency to the attached surface using the *Edit curve* option. The object coordinate system was activated for this step. Next, a sketch was designed in the coronal plane using the *New* command from the *Sketch* table. The preferred method was mid plane, the cell count of the sketch was set between 100 and 150, and the cell size was set at 1. In order to properly align and rotate the sketch from the coronal plane to the sagittal plane, the interactive *Translate* and *Rotate* options were used from the *Align* menu. Using the newly oriented sketch in the sagittal plane, the skull mesh was copied and mirrored, creating two overlapping objects in the current project. To perfectly overlap and align the two mesh skulls, the interactive translate and rotate commands were used again. To start creating the outline of the implant, two sketches were needed, thus a new sketch in the coronal plane was created. References were imported into the coronal sketch using the import references operation. The intersection is defined as the common points between the two skulls, the initial and the mirrored one. Next, the sketcher is used to project these intersection points into the coronal sketch. To create an outline of the skull profile into this sketch, a spline is defined using the *Create spline* command, which contains all intersection points created in the previous step. Based on the spline in the coronal plane, which outlines the contour of the mirrored skull, and on the initial 3D curve designed alongside the perimeter of the skull defect, a new surface is constructed using the *Surface construction* operation. Thus, the first surface of the bespoke cranial implant was created. This surface was separated from the other two objects and moved into a new part. Thickness was added to this surface to replicate the cranial bone in both shape and size. A number of measurements were undertaken on the initial skull part to establish the position of variable thickness alongside the perimeter of the defect. Boolean subtraction was done between the created solid implant and the wrapped skull to imprint the same shape on the edges of the implant. In order to remove undercuts from the implant edges chamfer, fillet and smoothing operations were performed.

After conducting all the above stages, the final characteristics of the three implants were as follows: Patient 1: perimeter—36.3 cm; area—21.775 cm^2^, volume—73.613 mm^3^, overall dimension bounding box—100.4408 mm × 82.3363 mm × 92.7498 mm;Patient 2: perimeter—29.3 cm; area—16.378 cm^2^, volume—34.53 mm^3^, overall dimension bounding box—58.8345 mm × 106.3967 mm × 84.9439 mm;Patient 3: perimeter—23.5 cm; area—11.697 cm^2^, volume—39.544 mm^3^, overall dimension bounding box—67.5562 mm × 101.0231 mm × 74.7505 mm.

The reconstruction stage resulted in six STL files (Figure 4), three of the bespoke implants and three of the patients’ preoperative damaged skulls, which were further studied to be manufactured using material extrusion (MEX) technologies at the premises of the Faculty of Industrial Engineering and Robotics from University POLITEHNICA of Bucharest. Meshes for all three implants were created with the same characteristics using a hexahedral eight-point element type. The voxel grouping had a 1 × 0.45 XY resolution and a 1 × 0.90 Z resolution. The implant for Patient 1 had a generated mesh with 308,924 triangles and 154,432 points. Patient 2 had a mesh with 492,726 triangles and 246,363 points. The mesh for Patient 3 was generated with 114,400 triangles and 57,206 points.

To obtain the best surface quality and mechanical characteristics of the 3D-printed parts, a parameter optimization analysis was undertaken. Five M300 Plus 3D printers were selected for the manufacture of all six STL files. Using Z-Suite software, some of the parameters were kept constant, while some of them varied in order to identify the best option for final manufacturing. Platform temperature was constant throughout the print jobs at 80 °C and all parts had seven layers of raft. The custom implants had a first layer gap of 0.31 mm and the defected skulls a value of 0.45 mm for this parameter. Maximum wall thickness was 3.13 mm for the entire study. Quality was set high with a normal print type, while the contour-infill gap had a value of 0.4 mm and a contour-top gap value of 0.25 mm. Due to individual stress scenarios, all implants were printed with a 90% infill parameter, while the skull parts had an infill of 40%. Eight top surface layers and four bottom surface layers were necessary for all parts. Three materials were selected for this study due to their mechanical properties, ease of printing, and availability, namely, polylactic acid (Z-PLA), acrylonitrile butadiene styrene copolymer (Z-ABS), and high impact polystyrene (Z-HIPS). The lowest layer height available was used in order to ensure a good surface finish for the implants, whilst a 0.29 mm layer height was used for the 3D printing of the skulls to maximize production time. Print pattern types were also varied to find the best time to quality ratio. Build orientation was the same for the skull STL files, maintaining the transversal (axial) plane parallel to the build plate of the manufacturing equipment. Implants were oriented with the inner concave surface away from the build plate, thus avoiding the positioning of support structures and potential damage to the surface quality. This surface was of particular functional importance as it comes in direct contact with the soft tissues surrounding the brain of the patient after cranioplasty is performed. Any unevenness or protuberance could lead to unwanted pressure points or swelling of the surrounding tissues. A support structure angle of 55 degrees was set for all prints and the *Smart bridges* and *Support lite* options were activated to ensure easy removal without extensive surface damage. Table 3 presents the parameter optimization analysis undertaken for the first patient.

Both Z-ABS and Z-HIPS allow for more precise manufacturing with a 0.09 mm layer thickness compared to Z-PLA, which can print at a minimum layer thickness of 0.14 mm. Although it is preferred to use one of the two aforementioned materials in order to obtain an accurate surface finish, it was important to have a balance between material consumption, costs, and printing time. Thus, results show that implant 1 is best fitted for Z-ABS manufacturing with PATT.3 pattern type, offering the best material consumption (110 g) from the 0.09 mm material set. At EUR 5.73, this option also has a competitive price with the Z-PLA group, which is the cheapest analyzed 3D-printing filament at EUR 33.32 per 800 g spool of material. As surface finish was not a target of the skull printing process, time and cost criteria were used to select the best option. Thus, skull was set to be 3D printed in Z-PLA with PATT.3 due to the best print time at 4 days 2 h and 09 min. This scenario also offers a comparable price (EUR 41.73) and material consumption (1002 g) with the other options in the same material group.

For the other two patients, analyses were done similarly and led to the selection of PLA as the main printing material for both defected skulls, with PATT.0 in case of Patient 2 and PATT.3 in Patient 3. The time estimate for skull 2 was 4 days 6 h and 8 min, with a material cost of EUR 47.80 and skull 3 was printed in 4 days 4 h and 53 min with a material cost of EUR 45.21. Implant 2 was set to be printed from HIPS at a 0.09 mm layer height in PATT.3 with a time estimate of 17 h and 32 min, whilst implant 3 obtained best values for ABS with 14 h and 12 min in PATT.0. A viable option to lower the 3D-printing time, but obtain the same fitting anatomical features, was to section the skull in 3-matic so as to obtain a smaller 3D model, which only contains the affected area [41].

Special attention was given to removing support structures from the convex surface of implants and the border surface of the skull defects, as not to create any cervices which did not reflect the anatomy of the patients. Sanding paper was used to remove any marks from support structures on the convex surfaces of the 3D-printed implants and ensure a smooth surface for the skin flap to sit on top of. In this stage, the surface finishing of the implants is key, because the PMMA final bespoke implants replicate identically the surfaces of the 3D-printed part. This is due to the fact that the liquid silicone rubber, from which the molds were made, fill and copy any imperfections of the master part [42].

Surface roughness was measured for all three implants to ensure an accurate surface finish (Table 4). A full report on the surface roughness is presented in Appendix A, which includes the corresponding roughness curves for each measurement.

Fit tests were undertaken for each 3D-printed implant and corresponding skull defect. In order to establish dimensional accuracy, measurements were undertaken on all three axes for the 3D-printed master part implants (Table 5). The obtained values are inside limit values recommended in the literature [43].

Measurements show a maximum difference of 0.6737 mm between the STL files and the 3D-printed parts on the Y axis of implant 1 and a minimum difference of 0.4095 mm on the Z axis of implant 3. Additional shaping of the edges with an electrical drill was done to accommodate a perfect assembly between the parts. Surface roughness and overall dimensions can be further improved through 3D-printing parameter optimization. Once the fit was validated, the surgeons were provided with the 3D-printed parts for surgical planning. Operating strategies, orientation, and fixing systems were tested and established during this phase, reducing the overall duration of the actual surgical procedure. This allowed the patients to be under anesthesia for a shorter amount of time, thus lowering the probability of further complications [44].

Following validation received from surgeons, bicomponent silicone molds were manufactured to allow intraoperative casting of the final PMMA implants. PMMA was chosen as the preferred material to manufacture the custom implant due to its advantages and potential improvements on limitations reported in the literature. PMMA has an elastic modulus of 3 GPa, which is higher than that of titanium, at 1.10 GPa [32]. Therefore, it can reduce stress shielding and loosening of fixation devices over time. With an impact strength of 5.27 kJ/m^2^, PMMA implants provide comparable impact strength to normal cranial bone. Moreover, PMMA exhibits superior compression and stress resistance compared to hydroxyapatite and has demonstrated its capability to adhere to the dura mater without any adverse reaction in the underlying tissue [45]. There are also specific requirements when working with PMMA. The production of PMMA involves a process that generates heat, which can reach temperatures of up to 80–100 °C for a duration of 5–8 min [46], thus direct contact with human tissue was avoided by in situ casting of the acrylic bone cement into the custom mold. The main supplies for manufacturing the molds were the master pattern, silicone rubber base, silicone rubber catalyst, electronic scale, mixing recipient, demolding spray, modeling clay, and acrylic bone cement kit (Figure 5).

ZA 22 has a medium elasticity with a Shore A of 22, which allowed the demolding of the bone cement implant without any breakage. It was also firm enough so that it did not change shape when pressure was applied to form the PMMA test implant.

### 3.2. Intraoperative Actions

To ensure the safety and success of the cranioplasty, intraoperative surgical theatre preparation was highly important.

The surgical team scrubbed in and donned sterile surgical gowns and gloves to prevent the introduction of bacteria and other contaminants into the surgical site. Patients were prepped and positioned in a way that allowed access to the surgical site while ensuring their comfort and safety during the procedure. Next, the surgical field was prepared by cleaning the area with an antiseptic solution, such as iodine or chlorhexidine, to reduce the risk of infection. The surgical field was draped with sterile surgical drapes to create a barrier between the surgical site and non-sterile areas of the operating room. Anesthesia was administered to each patient, in correlation to personal medical data and pre-existing conditions, to ensure their comfort and safety during the procedure. Once the patients were properly anesthetized, the surgeon made an incision alongside the cranial defect of each patient to access the surgical site and performed the cranioplasty procedure. Visual inspection of the cranial defect was performed and the final PMMA custom cranial implants were test fitted on the real affected skull area of patients.

Fixation of the PMMA bespoke implants was done using the CranioFix 2 Titanium Clamp System (Figure 6) due to the short time frame required to put it in place. It also offered maximum flap stability in relation to the cranial edge. The first patient was fitted with five titanium 11 mm FF490T clamps, due to the large cranial damaged area (approximately 36 cm in perimeter). The second patient had the PMMA implant fixed with four identical titanium clamps. Patient number three required extra support due to the very irregular edges and was fitted with two 11 mm FF490T titanium clamps and three linear titanium cranial plates and screws.

After the cranioplasty procedure, wound closure and site sterilization were important steps in promoting healing and preventing infection. Skin and subcutaneous tissue incisions were sutured using non-absorbable polyethylene and silk wires. After the wound was closed, the surgical team applied an antiseptic solution to the skin around the incision site to help prevent infection. This included using a sterile swab to apply an iodine-based solution. After the procedures were complete, the surgical team worked to ensure the patients’ comfort and safety during recovery, including monitoring for any signs of infection or complications.

### 3.3. Postoperative Actions

Possible complications of cranioplasties can be registered immediately after surgery or even years after the procedure [47]. The surgical teams evaluated all patients to catch early signs of short-term complications, such as infection, bleeding, swelling, hematoma, seizures, and even neurological complications [48]. In order to lower the risks or even avoid short-term complications, each patient was prescribed a custom medication plan in correspondence with their medical history and underlying medical conditions. Generally, the plan included antibiotics, anticoagulants, and anti-inflammatory and pain medication. Patients spent 24 h in the intensive care unit (ICU) of the hospital under permanent monitoring of vital signs, including heart rate, blood pressure, and oxygen levels, to ensure that they remained stable. Patients did not develop any infections, hematomas, or seizures. Brain damage and strokes were also avoided for all three patients. After stabilization and avoidance of short-term complications, patients were transferred to the neurosurgery department. Once the wound sutures were removed, the patients were released home with a specific monitoring and recovery plan, which aimed at lowering the risks of long-term complications. These can include implant failure, infection, chronic headaches, implant rejection, and cosmetic issues [49].

The monitoring and recovery plan included pain management, wound care, nutrition, physical therapy, and follow-up appointments. Pain management was an important aspect of postoperative care, aiding in patient compliance and overall comfort throughout the surgical experience. The overall goal was to improve the patients’ quality of life compared to before the surgery. Large craniectomies often leave the patient with restricted access to certain activities, which usually leads to isolation and depression [50]. Speeding up the recovery process through pain management and reintroduction into a normal lifestyle was essential for the final success of the procedures. Proper wound care was vital to prevent infection and promote healing. Nutrition was also a core factor for healing after the extensive surgeries. Patients were encouraged to eat a balanced diet and receive nutritional support. Physical therapy was needed to regain strength and mobility after a complex surgery. The patients attended follow-up appointments with the surgeons and medical team to monitor their progress and ensure that the recovery process was going smoothly. Postoperative CT scans were conducted for Patient 2 and 3 and an MRI was undertaken for Patient 1 (Table 6). CT image sets were captured at 1.25 mm and 1.5 mm slice thickness. Pixel spacing was set between 0.35 mm and 0.45 mm.

The CT characteristics were further used to import and process the DICOM image sets for each of the three patients in order to validate the implant positioning and patient outcome (Figure 7). According to the MRI scans, there were no detectable infections, inflammation, or swelling of soft tissues surrounding the implant.

Visual aesthetics validation was done immediately after wound suture but also in follow-up appointments (Figure 8). Symmetry and patient compliance were achieved in all three cases. During the follow up meetings, the patients did not report any complications. None of them have required revision surgeries so far, developed infections, or suffered implant failure or rejection. Long-term recurrent monitoring using MRI scans was recommended for all three patients.

During consultations, the medical team observed significant benefits and improvements for the three patients who underwent cranioplasties:Conducted cranioplasties helped protect the brain from injury and trauma by restoring the missing portion of the skull for all three patients.Appearance of the head and face was improved, which had a positive impact on the patients’ self-esteem and overall compliance.All three cranioplasties helped restored normal brain function, including cognitive function, motor function, and sensory function.The risk of infection was reduced by closing off the open space in the skull of the patients (all of them had previous craniectomies performed), which could have been a breeding ground for bacteria and other pathogens.All cranioplasties improved the patients’ quality of life by reducing pain, discomfort, and other symptoms associated with their previous craniectomies.Restoring the skull helped improve the mental health of the three patients by reducing anxiety and depression associated with both visible deformities or asymmetry, and with isolation from the lack of accessibility to certain activities due to safety risks.

### 3.4. Limitations and Future Development

The proposed integrative surgery management system can be applied on non-urgent stable patients with a previously performed craniectomy. The studied approach has been developed by the authors to implement the first two components (pre- and intra-operative) in a time frame of five days. Critical care patients often need medical intervention in a shorter period of time. However, cranioplasties are rarely performed together with craniectomies, patients undergoing the latter procedure needing between 6 weeks [51] and up to a year [46,47,48] to reach a stable medical state.

Another limitation of the study and of overall custom-made cranial reconstruction available techniques remains the generation of anatomical geometry, which is time-consuming and requires high skills of computer design or programming. Recently, advances have been made in streamlining this step through deep learning programs, artificial intelligence applications, or extrapolation CAD methods [52,53,54]. Nevertheless, these are still exploratory research that are yet to be fully available for patients worldwide. Future developments include automatization of the bespoke implant reconstruction using advanced image processing and face recognition algorithms. Another development avenue is the design of a fully parametric cranial implant based on surface feature recognition with input from patient medical data. Authors have developed similar customizable and adaptable cranial implants [55,56], but the design process of these medical devices is yet to be fully parametric.

Overall, the benefits brought by the ISMS to the studied patients outweigh the limitations presented above. Feedback from all involved participant parties (surgeons and medical professionals, engineers and technicians, and patients and family members) is key to the continuous improvement of the management system, with the patients’ well-being as a primary focus.

## 4. Conclusions

Traumatic brain injury (TBI) is a significant global health concern, with a high incidence of death and disability. TBIs often require a craniectomy followed by a cranioplasty, which in recent years have been undertaken using custom-made and additive manufactured bespoke implants. These implants are designed specifically for the patient using advanced imaging techniques, such as CT and MRI scans. The use of bespoke implants ensures a better fit and reduces the risk of complications. However, bespoke implants can be costly and time consuming to manufacture. This has led to a search for cost-effective solutions to make the technology more accessible to patients in low- and middle-income countries. The current paper proposes a study on developing and implementing an integrative surgery management system for cranial reconstructions using bespoke implants as an accessible and cost-effective solution. The study aimed to design, manufacture, and implement bespoke cranial implants using readily available tools and materials, such as standardized and regulated bone cement materials. The goal was to lower costs, reduce intraoperative times, and improve patient outcomes.

Three patients were selected for the study, all of whom required cranioplasty surgery following a craniectomy due to severe TBI. The patients underwent preoperative imaging scans to generate 3D models of their skulls, which were used to design bespoke cranial implants. The implants were then manufactured using a combination of 3D printing and standard bone cement materials. All three patients underwent successful cranioplasty surgeries using bespoke implants. Postoperative evaluations showed improvements in patient compliance and overall quality of life, with no complications registered from both short-term and long-term monitoring. The bespoke implants provided a better fit than traditional metal 3D printed implants, resulting in increased patient satisfaction.

The study demonstrated that an integrative surgery management system for cranial reconstructions using bespoke implants is a cost-effective and accessible solution for patients in low- and middle-income countries. The use of standardized and regulated bone cement materials for the manufacture of bespoke cranial implants lowered costs compared to metal 3D printed implants. The pre-planning management stages reduced intraoperative times, leading to a better implant fit and overall patient satisfaction.

In conclusion, the use of bespoke cranial implants is a significant advancement in the field of cranioplasty surgery. However, the high-cost and time-consuming manufacturing process can make the technology inaccessible to patients in low- and middle-income countries. This study provides evidence that an integrative surgery management system for cranial reconstructions using bespoke implants is a cost-effective and accessible solution for patients in these countries. This technology can help to improve patient outcomes and reduce the incidence of TBI-related deaths and disability worldwide.

## Figures and Tables

**Figure 1 bioengineering-10-00544-f001:**
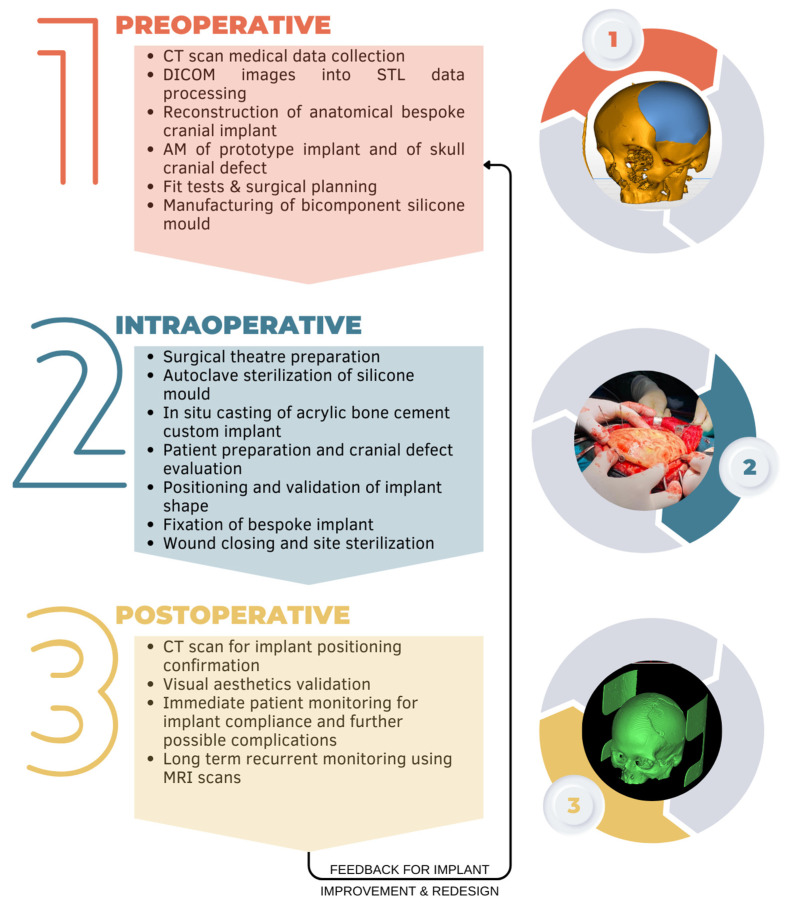
Integrative surgery management system for cranial reconstructions using bespoke implants.

**Figure 2 bioengineering-10-00544-f002:**
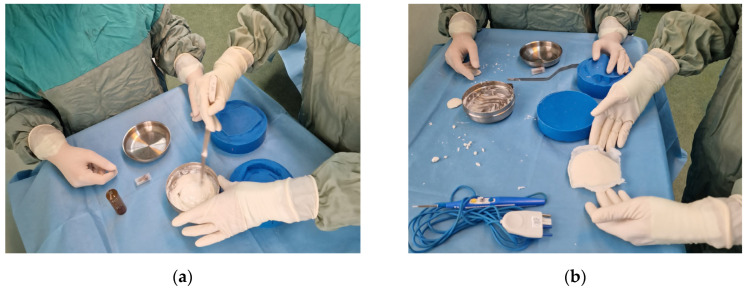
Intraoperative manufacturing of bespoke implant for cranial reconstruction: (**a**) PMMA activation and (**b**) demolded bespoke cranial implant.

**Figure 3 bioengineering-10-00544-f003:**
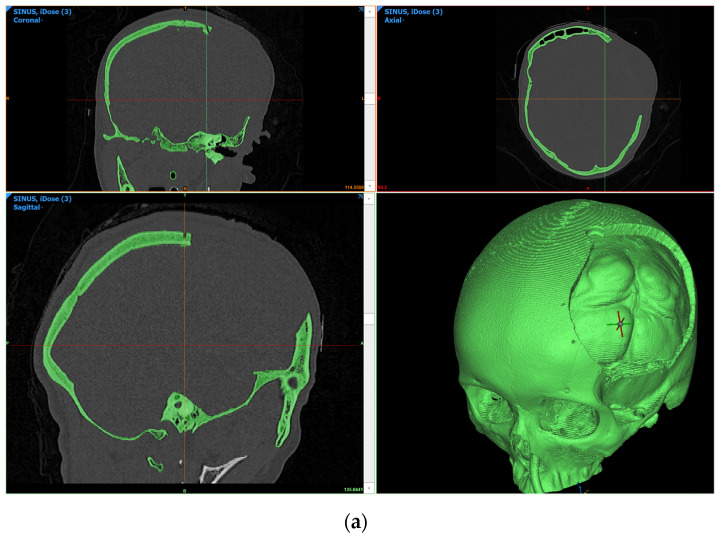
CT bone masks form preoperative CT scans: (**a**) medical data of Patient 1; (**b**) medical data of Patient 2; and (**c**) medical data of Patient 3.

**Figure 4 bioengineering-10-00544-f004:**
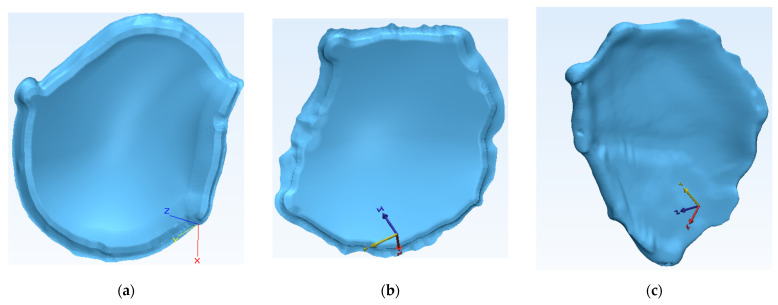
Geometries of all reconstructed cranial implants: (**a**) custom implant for Patient 1; (**b**) custom implant for Patient 2; and (**c**) custom implant for Patient 3.

**Figure 5 bioengineering-10-00544-f005:**
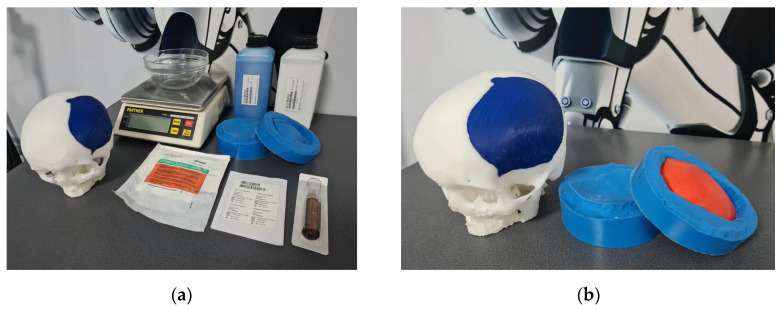
Manufacturing, assembly fit, and test of silicone mold and prototype implants: (**a**) materials for manufacturing of silicone mold and bone cement for validation and (**b**) silicone mold fitted with 3D-printed implant and bio-model of skull.

**Figure 6 bioengineering-10-00544-f006:**
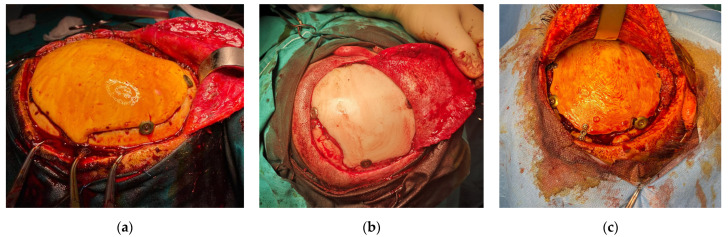
Intraoperative implant validation and fixation: (**a**) cranial implant for Patient 1; (**b**) cranial implant for Patient 2; and (**c**) cranial implant for Patient 3.

**Figure 7 bioengineering-10-00544-f007:**
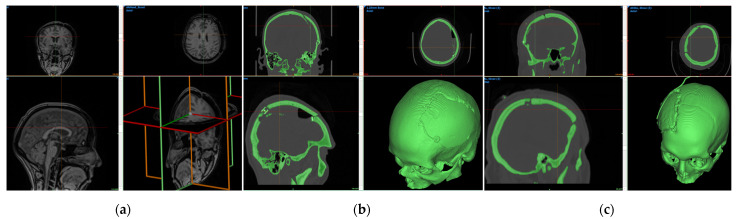
Postoperative medical data after scan processing: (**a**) MRI scans of Patient 1; (**b**) CT bone mask from medical data of Patient 2; and (**c**) CT bone mask from medical data of Patient 3.

**Figure 8 bioengineering-10-00544-f008:**
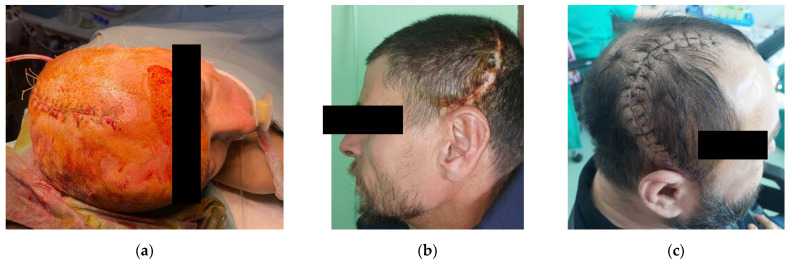
Postoperative visual aesthetics validation: (**a**) immediate postoperative of Patient 1; (**b**) one month postoperative of Patient 2; and (**c**) two weeks postoperative of Patient 3.

**Table 1 bioengineering-10-00544-t001:** Characteristics of preoperative CT scans.

Tag Description	Value		
	Patient 1	Patient 2	Patient 3
Institution Name	SUUMC_iCT256	SUUMC_CT128_UPU	SUUMC_CT128_UPU
Patient’s Sex	F	M	M
Patient’s Age	38	47	44
Manufacturer	Philips	GE MEDICAL SYSTEMS	GE MEDICAL SYSTEMS
Manufacturer’s Model	iCT 256	Revolution EVO	Revolution EVO
Study Date	25 September 2020	29 June 2022	30 January 2023
Modality	CT	CT	CT
Study Description	CAP STD.	CER	HEAD
Series Description	SINUS, iDose (3)	CT IAC4 Sag 0.6 Avg	1.25 mm STD
Image Type	ORIGINAL\ PRIMARY\AXIAL	DERIVED\ SECONDAY\ REFORMATTED\ AVERAGE	ORIGINAL\PRIMARY\AXIAL
No. of Images	200	547	155
Slice Thickness	0.9 mm	0.625 mm	1.25 mm
Software Version	4.1	-	cj_digital.46
Protocol Name	CAP STD./Head	1.1 Cerebral Helical	1.1 Cerebral Helical
Pixel Spacing	0.44921875\0.44921875	0.488281\0.488281	0.525391\0.525391

**Table 2 bioengineering-10-00544-t002:** Mechanical characteristics of potential materials involved in the manufacturing process.

Mechanical Characteristics	Implant		Master Part		
	PMMA	Ti	Z-PLA	Z-ABS	Z-HIPS
Tensile Strength	75 MPa	140 Mpa	47.95 Mpa	30.46 Mpa	16.90 Mpa
Elongation at Break	4.5%	10%	4.32%	11.08%	7.75%
Flexural Modulus	3 Gpa	1.1 Gpa	1.47 Gpa	1.08 Gpa	1.18 Gpa
Izod Impact, Notched	5.27 kJ/m^2^	22 kJ/m^2^	3.14 kJ/m^2^	8.93 kJ/m^2^	4.82 kJ/m^2^

**Table 3 bioengineering-10-00544-t003:** 3D-printing parameter optimization for the manufacture of the bespoke cranial implant and defected skull for Patient 1.

3D-Printed Part	Material	Print Pattern	Estimated Print Time (d h min)	Material Usage [m]/[g]	Material Cost (EUR) ^1^
Implant 1	Z-PLA (0.14 mm layer height)	PATT. 0	16 h 00 min	47.41 m/135 g	5.62
		PATT. 1	18 h 09 min	46.68 m/133 g	5.54
		PATT. 2	19 h 26 min	50.95 m/145 g	6.04
		PATT. 3	16 h 59 min	45.15 m/129 g	5.37
	Z-ABS (0.09 mm layer height)	PATT. 0	19 h 12 min	49.28 m/117 g	6.09
		PATT. 1	23 h 13 min	48.48 m/115 g	5.99
		PATT. 2	1 d 2 h 09 min	54.16 m/129 g	6.72
		**PATT. 3**	**21 h 18 min**	**46.09 m/110 g**	**5.73**
	Z-HIPS (0.09 mm layer height)	PATT. 0	19 h 50 min	48.46 m/120 g	7.85
		PATT. 1	23 h 51 min	47.66 m/118 g	7.72
		PATT. 2	1 d 2 h 45 min	52.59 m/130 g	8.51
		PATT. 3	21 h 56 min	45.09 m/112 g	7.33
Skull 1 (0.29 mm layer height)	Z-PLA	PATT. 0	4 d 2 h 10 min	353.36 m/1007 g	41.94
		PATT. 1	4 d 3 h 32 min	350.14 m/998 g	41.57
		PATT. 2	4 d 2 h 53 min	356.01 m/1015 g	42.27
		**PATT. 3**	**4 d 2 h 09 min**	**351.40 m/1002 g**	**41.73**
	Z-ABS	PATT. 0	4 d 2 h 34 min	349.75 m/832 g	43.32
		PATT. 1	4 d 3 h 52 min	346.61 m/825 g	42.95
		PATT. 2	4 d 3 h 22 min	352.19 m/838 g	43.63
		PATT. 3	4 d 2 h 39 min	347.93 m/828 g	43.11
	Z-HIPS	PATT. 0	4 d 0 h 08 min	345.71 m/856 g	56.03
		PATT. 1	4 d 1 h 28 min	342.70 m/848 g	55.50
		PATT. 2	4 d 0 h 55 min	348.07 m/862 g	56.42
		PATT. 3	4 d 0 h 12 min	343.95 m/851 g	55.70

^1^ Costs were estimated from https://store.zortrax.com/ (accessed on 18 March 2023) for 800 g material spools of standard materials: Z-PLA—33.32 €, Z-ABS—41.65 €, and Z-HIPS—52.36 €.

**Table 4 bioengineering-10-00544-t004:** Surface roughness measurements.

Point No.	Implant 1			Implant 2			Implant 3		
	Ra [μm]	Rz [μm]	Rq [μm]	Ra [μm]	Rz [μm]	Rq [μm]	Ra [μm]	Rz [μm]	Rq [μm]
1	2.400	12.921	2.997	2.886	14.525	3.539	7.227	32.905	8.755
2	2.209	9.934	2.747	4.575	21.106	5.459	7.865	36.636	9.470
3	4.841	22.072	5.831	4.947	22.584	6.027	6.558	28.141	7.768
4	6.399	32.335	8.225	6.220	26.595	7.431	9.718	39.651	11.600
5	5.990	27.175	7.611	6.211	29.011	7.551	6.995	32.210	8.853
6	3.048	15.771	3.814	6.546	30.151	7.925	11.074	46.386	13.279
7	10.991	46.850	13.420	10.109	41.110	11.632	8.546	34.906	10.152
8	3.644	18.313	4.565	8.362	35.679	9.934	13.056	49.595	15.078

**Table 5 bioengineering-10-00544-t005:** Dimensional accuracy of 3D-printed implants compared with the CAD files.

Axis	Implant 1 Dimensions [mm]		Implant 2 Dimensions [mm]		Implant 3 Dimensions [mm]	
	STL	Measured	STL	Measured	STL	Measured
X	100.4408	100.92	58.8345	59.47	67.5562	68.09
Y	82.3363	83.01	106.3967	106.98	101.0231	101.65
Z	92.7498	93.24	84.9439	85.52	74.7505	75.16

**Table 6 bioengineering-10-00544-t006:** Characteristics of postoperative scans.

Tag Description	Value		
	Patient 1	Patient 2	Patient 3
Institution Name	Policlinica Constanta	SUUMC_CT128	SUUMC_iCT256
Patient’s Sex	F	M	M
Patient’s Age	38	47	44
Manufacturer	SIEMENS	GE MEDICAL SYSTEMS	Philips
Manufacturer’s Model	Sempra	Revolution EVO	iCT 256
Study Date	4 July 2022	14 November 2022	9 March 2023
Modality	MR	CT	CT
Study Description	CAP NORMAL^STANDARD + SPACE	CEREBRAL	CEREBRAL
Series Description	tof_fl3d_tra_p2_multi-slab	1.25 mm Bone	OS, iDose (3)
Image Type	ORIGINAL\PRIMARY\M\ND\NORM	ORIGINAL\PRIMARY\AXIAL	DERIVED\ SECONDAY\MPR
No. of Images	139	140	155
Slice Thickness	0.6999	1.25	1.5
Software Version	syngo MR E11	cj_kl.89	4.1
Protocol Name	tof_fl3d_tra_p2_multi-slab	1.1 Cerebral Helical	CAP STD./Head
Pixel Spacing	0.3515625\0.3515625	0.449219\0.449219	0.4014085/0.4014085

## Data Availability

The data presented in this study are available on request from the corresponding author.

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
