# Peer review of "Bespoke Implants for Cranial Reconstructions: Preoperative to Postoperative Surgery Management System"

_bioengineering, 2023, doi:10.3390/bioengineering10050544_

Round 1

Reviewer 1 Report

Review report on the paper " bioengineering-2347019" submitted for publication in Bioengineering Journal

The manuscript reports preoperative to 2 postoperative surgery management systems of 3D-printed bespoke implants for cranial reconstructions.

i)             The manuscript has been written well and organized; however, the abstract should be modified with more precise details, specifically, line 32; “costs……implant” comparison with what?

The manuscript may be publishable. However, since the manuscript contains more clinical relevance than the 3D printing details, I would highly recommend considering other reviewer’s comments, who are experts in the clinical field.

It would be nice if a minor English correction made 

Reviewer 2 Report

The paper entitled “ Bespoke Implants for cranial reconstructions: preoperative to postoperative surgery management systemfocuses on an integrative surgery management system for developing bespoke implants used in cranial reconstructions with the final target of improving patient outcomes and well-being, through the reduction of the main limitations of bespoke cranial implants. The paper is interesting and well-explained. Although the introduction refers to the aim of the study and the results are understandably submitted, the arrangement of the text in the study should be changed.

I would like to recommend the publication of the manuscript in this journal after fulfilling the following recommendations:

1.     First of all, there is a tendency to mix the methodological section with the results. The model of the FDM printers and autoclaves should be pointed out in section 2.

2.     The comparison between the mechanical characteristics of PMMA, Ti, as well as Z-PLA, Z-ABS, Z-HIPS should be given in Table(s) in section 2. The reasoning for choosing PMMA should be shifted to section 3 Results and Discussions.

3.     Table 1 should be shifted to section 2 since it is related to the methodology of the work.

4.     After Figure 2, the geometries of all printed implants should be shown as figures. What meshes and model positioning were found to be the most suitable?

5.     The methodology of casting, procedures of sterilizing, autoclaving, cleaning, fitting, fixation, etc. should also be shifted to section 2.

6.     The authors mentioned that “Fit tests were undertaken for each 3D printed implant and corresponding skull defect.”. Then, what is the accuracy of all printed implants when compared to the STL files?

7.     The authors stated that “…it is preferred to use one of the two aforementioned materials in order to obtain an accurate surface finish…” but do not present measurements of the surface roughness of the printed parts. Optimizing this parameter will eliminate the need for grinding/polishing and will reduce manipulation time.

8.     Since PMMA is a 3D printable material, why should the authors use Z-PLA, Z-ABS, Z-HIPS for printing and then casting PMMA implant?  To reduce intraoperative times a solution for direct PMMA printing should be sought.

Minor corrections to the English grammar and style are needed. 

Round 2

Reviewer 2 Report

The authors have carefully addressed the reviewer's recommendations. One more thing - the text from lines 392-408, 456-469, and 478-508 concerning 3D printing technique, mold manufacturing, and surgical theatre preparation should also be shifted to section 2. 

The English grammar and style are fine. 
